# The Hydrophobic Effect Studied by Using Interacting Colloidal Suspensions

**DOI:** 10.3390/ijms24032003

**Published:** 2023-01-19

**Authors:** Francesco Mallamace, Giuseppe Mensitieri, Martina Salzano de Luna, Domenico Mallamace

**Affiliations:** 1MIFT Department, Istituto Sistemi Complessi del CNR, University of Messina, 00185 Rome, Italy; 2Department of Chemical, Materials and Production Engineering, University of Naples Federico II, Piazzale Tecchio 80, 80125 Naples, Italy; 3Departments of ChiBioFarAm, Section of Industrial Chemistry, University of Messina, CASPE-INSTM, V.le F. Stagno d’Alcontres 31, 98166 Messina, Italy

**Keywords:** water, hydrophobic effect, local order, relaxation times, self-diffusion

## Abstract

Interactions between nanoparticles (NPs) determine their self-organization and dynamic processes. In these systems, a quantitative description of the interparticle forces is complicated by the presence of the hydrophobic effect (HE), treatable only qualitatively, and due to the competition between the hydrophobic and hydrophilic forces. Recently, instead, a sort of crossover of HE from hydrophilic to hydrophobic has been experimentally observed on a local scale, by increasing the temperature, in pure confined water and studying the occurrence of this crossover in different water–methanol solutions. Starting from these results, we then considered the idea of studying this process in different nanoparticle solutions. By using photon correlation spectroscopy (PCS) experiments on dendrimer with OH terminal groups (dissolved in water and methanol, respectively), we show the existence of this hydrophobic–hydrophilic crossover with a well defined temperature and nanoparticle volume fraction dependence. In this frame, we have used the mode coupling theory extended model to evaluate the measured time-dependent density correlation functions (ISFs). In this context we will, therefore, show how the measured spectra are strongly dependent on the specificity of the interactions between the particles in solution. The observed transition demonstrates that just the HE, depending sensitively on the system thermodynamics, determines the hydrophobic and hydrophilic interaction properties of the studied nanostructures surface.

## 1. Introduction

Water, despite being an apparently simple molecule, is instead characterized by complex thermodynamic anomalies [1,2,3,4,5]. As it is well known, these complexities are due to the intermolecular hydrogen bond (HB) interaction, which governs the system structure, giving rise to the tetrahedral local order [1]. In ordinary ice, and in the liquid state, each water molecule has four nearest neighbors, acting as a hydrogen donor to two of them and another receptor from the other two. In the liquid, these tetrahedral water patches occur at $T^{*}≃320$ K and decrease as *T* increase progressively, with a specific volume larger than the overall average [6]. Thus, the entropy always decreases with *T* and the specific heat is, of necessity, positive with large fluctuations in specific volume, entropy, and their negative cross-correlations. HB is also the basis of hydrophilicity with other molecules [7,8,9,10]. This HB local order gives rise to water polymorphism observed first in the amorphous ice [11] of then in the liquid state [1,7,8,10]. It consists in the simultaneous presence of two phases of different densities: LDA and HDA (the low and high density amorphous, respectively) and the low-density liquid (LDL) and high-density liquid (HDL) [1,8,10]. All the water models are based on this fundamental property confirmed by experiments in confined water and simulations in bulk water [12,13,14,15].

A property, exactly opposite to hydrophilicity, is hydrophobicity: the way which some chemical species “refuse” any interaction with water. Moieties with these properties determine properties of organic compounds, such as amphiphilic molecules [16]. Nowadays, a reasonable understanding of the hydrophobic interactions is far to be achieved; unfortunately, we do not even have any analytical form to treat it quantitatively [17,18]. In many materials, both of these interactions are the keys to understanding their properties [19], since their competition, at the origin of molecular configurational variations, is known as the entropic-dependent hydrophobic effect (HE). The resulting conformation in the hydration shells around a hydrophobic solute can affect, as suggested by MD simulation studies [20], the structure of the solute itself.

Originally, HE was described in terms of a wetting process, i.e., the macroscopic behavior of the surface tension, a *T*-dependent quantity [21], characteristic of liquids (hydrophobic, hydrophilic and their mixtures). In liquid solutions, due to the HE phenomenon, self-aggregation originate, and changes in *T* and composition (*X*), these lead to dramatic thermodynamic differences. In addition, at the lowest *T* (or at high (*X*)), the process of the liquid glass transition (GT) also takes place, with divergent growth of the transport functions (e.g., the relaxation times). Hence, such a HE description is no longer valid with regards to being relevant to the nanostructure interactions and how they are mediated by the solvent. Nowadays, the HE is considered the scale-dependent manifestation of a segregation, occurring in water–hydrophobic systems, dependent on the way in which the hydrophobic molecules, individually hydrated, interact and self-assemble [22,23]. Despite the hydrophobic role in self-assembly processes over multiple length scales (from the molecules to mesoscopic aggregates), the HE is only qualitatively described [23,24,25]. The interaction of hydrophobic substances with water reveals subtle and intriguing situations depending on their sizes, chemistry, and thermodynamics [23].

The length–scale dependence of hydrophobicity, with some universality aspects, constitutes an intriguing issue in material sciences. The small and large solutes hydration difference seems to arise from the way in which they affect the water structure [17,26,27]. The fluctuating water networks can accommodate a small hydrophobic solute without HB changes, but many HBs are broken in the case a large solute [23]. Hence, the solvation and self-assembling of large structures takes place, originating nanostructures of different geometry (from spherical to layered) in processes sensitive to thermodynamic variations. These structures, and their dynamics can be described in terms of current statistical physics by using scaling laws and universality concepts [28,29].

Water in small solute determines a relevant entropic penalty, which comes from its restricted configurations in the solute’s shell. When solutes associate, some of these restricted water molecules are released by changing their entropy, and the molecular assembly driving force increases with *T*. Experiments show the universality in the length–scale dependence of hydrophobicity by testing a surface water depletion layer (less than the one molecule dimension for the large fluctuations in the interface frustrated water) [17,30]. This universality is the connection point between the large and small scale hydrophobicity, and the hydrophobic solute self-assembling can be used to interpret the large scale HE behavior. For water confined in a hydrophobic bilayer, the entropy is determinant [18,31]: when the bilayer gap is critically small, water is ejected, in contrast to stable water between hydrophilic films [30]. Other models propose that, for the HE, a key concept is the hydrophobic object size [32,33]. Water can wrap efficiently around hydrophobic elements with a radius of curvature of molecular size. If it meets flatter hydrophobic surfaces, it forms a molecularly thin cushion of depleted density.

Precise insights into the hydrophobicity origins are not easy to ascertain: it is hard to probe hydrophobic solutes experimentally. Only few techniques have a proper resolution to focus in a detailed way the hydrophobic molecules and water interactions. Therefore, between the four basic interactions—the electrostatic, the van der Waals the hydrophobic, and the HB potentials—only the hydrophobic one is not known [18,22,34,35].

The same holds for nanoparticles (NPs) resulting from the self-aggregations. Multiscale collective processes can account for their interactions, but a quantitative description encounters severe obstacles. The reason is in their (experimentally proven) non-additivity, making impossible the decomposition of the resulting mean force (NPMF) potential between NPs into separate contributions [35]. New experiments able to examine interactions at the nanoscale should help to provide new details on these force fields. An example can be the surfactant solutions, where the HE is the relevant NPMF contribution. Hydrophobic interactions, creating structures, are stabilized by increasing entropy rather than enthalpy [36]. The same happens in bio materials, where the HE forces drive the design of peptide amphiphiles. Multiplicity and non-additivity suggest, thus, that NPs, being unique in their interactions, can constitute a correct pathway to clarify the of HB and HE contrasting roles.

HE and molecular association, from short to large tail molecules, has been studied by using several different experimental techniques [37,38,39,40,41,42,43,44,45] focused on their structure and dynamics, but, despite these many efforts, we are far from a clear understanding of the basic processes at the HE origin. Recent models and experimental findings suggest that the appropriate way to describe NPs’ basic properties is the use of a statistical approach on collective properties with proper energy configurations, rather than the “simple identify” of replicated objects (molecule) [46].

An interesting finding on the HE comes out from a NMR study conducted in water confined in hydrophobic nanotubes that switches from hydrophobic to hydrophilic when the *T* is lowered from 295 to 281 K [47]. A slowdown in molecular reorientation of such adsorbed water was detected, probing that the water structure has intriguing *T* dependences. This is all in line with the predictions that such a crossover can happen because the free energy of a full nanotube is very close to that of an empty one [25].

Similar studies made in water/methanol solutions (at different molar fractions (*X*) in a wide *T*-range, 200–330 K) have shown mutual hydrophobic–hydrophilic effects on the molecular dynamics [48]. This occurs through the simultaneous study of the molecular groups (the water (OH*_W_*), methanol (OH*_M_*) hydroxyls, and the methanol methyl group (CH_3_, the only hydrophobic moiety present. The corresponding correlation times $\tau_{c}$ are thus measured, related to the spin–spin interaction, and related to the system T, *X* evolution. The main observation is that, in the water supercooled regime, these correlations are stronger with respect to ambient because the HB interactions have a lifetime long enough to sustain a stable water network. However, when increasing *T*, it progressively decreases because HE destroys this clustering with a consequent decoupling in the system dynamic modes. For T>265 K, the hydrophobicity becomes gradually stronger and governs the physical properties of the solutions, giving rise to a sort of hydrophilic–hydrophilic transition. Of special interest was also the limit of pure bulk water (X=0), which shows around this crossover a marked decrease in its hydrophilic interaction, in analogy with the confined one becoming hydrophobic for T>285 K [47].

Starting from these considerations, and essentially from the non-additivity of NPs basic interactions, we have considered to give, in the present work, a new contribution to the HE investigation and on the reciprocal influence with hydrophilicity by using a photon correlation spectroscopy (PCS or quasi-elastic light scattering QELS are techniques which operate in the time domain [49]) from the evaluation of the relaxation dynamics of nanoparticle solutions of diverse solutes at different *T* and volume fraction φ. As we will see later, the basic idea of this work is to verify the different effects of the hydrophilicity and the hydrophobicity on a system of interacting nanoparticles. In other words, we will try to quantify the effects of HE on collective thermodynamic properties, rather than local and atomic, as proposed by the NMR technique.

The PCS is a scattering technique that allows the study of collective transport phenomena on mesoscopic scales. It describes the scattering of laser light from fluid media, where the presence of any density fluctuations will produce the Rayleigh scattering of incident light (broaded by the fluctuations time dependence). In such a way, the profile of this line mirrors the power spectrum of fluctuation and is related to the dynamic structure factor S(q,t) and thus to the system space–time correlations. The PCS focuses well many important collective phenomena of physical, chemical, and biological interest characterized by slow fluctuations in the range $10^{7}$ down to 1 s^−1^. Examples of this are the diffusion of macromolecules and macromolecular aggregates in solution and the fluctuations of density and concentration in one- and two-component fluids near the critical point or the glass transition.

These latter phenomena, together with the processes of self-aggregation and percolation, form the basis of modern statistical physics and are characterized by universal behavior and precise scaling laws. Scaling laws represent an accurate and elegant way of describing the functional interrelationships between two physical quantities that scale with each other over a significant interval. Historically, the use of scaling laws was born in the first model of aggregation and cluster formation (percolation) introduced a long time ago by the Flory-Stockmayer model [29]; it has been then widely and effectively used to describe the behavior of thermodynamic functions at phase transitions and their evolution at critical points [50]. Finally, the approach to the glass transition has been described in these terms (universality and scaling) by mode coupling theory (MCT) [51]. A singularity of kinetic origin—not related to thermodynamic singularities as in critical processes—is the most important prediction of the theory [52]. One of the merits of MCT is to identify the universal features of the time decay of density correlations, on approaching this singularity, in terms of asymptotic power laws. The corresponding physical interpretation is essentially related to a cage effect, i.e., the difficulty of a particle to move due to the crowd of the surrounding ones.

The nanoparticle we studied is a dendrimer (or starburst polymers) in methanol (and in water). These nanoparticles, synthesized by repeating units in a self-similar fashion, have, in our case, OH terminal groups. The actual PCS study was carried out for both the solvents by increasing φ up to the values of the dynamic arrest (DA), typical of supercooled liquids and glass forming materials [52]. In the case of methanol, we have also studied the system evolution also as a function of *T* (whereas for water as solvent we have considered only T=293 K). The reason is twofold, on the one hand, studying such a system in these *T* and φ ranges means to monitor the HE; on the other hand, the DA occurs at high concentrations and in accordance with mode coupling theory (MCT) with many significant differences if the used colloids interact as hard spheres or have an attractive interaction [52].

## 2. Results and Discussion

The next figures illustrate the ISF functions behavior for the methanol colloidal solutions at several diffrent values of φ as *T* decreases. Figure 1 deals with T=308 and 303 K. In the first case, the behavior is similar to that shown at T=313 K (figure in Section 3.2). None of the ISFs illustrated display the AHS logarithmic decay, whereas, in the second case (303 K), such a decay can be observed only for φ=0.37. In both cases, there is the suggestion that $\varphi_{c}>0.5$. It can also be observed that, for φ=0.27, the behavior of the corresponding ISF is that of a Brownian colloid. A subsequent *T* decrement leads to an increase in the concentration range in which the logarithmic decay is clearly visible.

From Figure 2, which show the different ISFs at the different values of φ, for T=298 K and 293 K, such a decay typical of AHS systems can be observable in the first case for 0.37<φ<0.39, whereas for T=298 K, it is in the range 0.37<φ<0.42. For higher values of φ, the IFS behavior is that typical of HS repulsive colloids; even in these cases, we have $\varphi_{c}>0.5$.

The lowest two temperatures (T=288 and 283 K), reported in Figure 3, instead propose a very interesting and different IFS behavior with respect to the one previously shown.

From what can be observed, at 283 K, it is evident that the logarithmic decay is present in all the liquid state up to the critical concentration, thus indicating that, at this temperature, the system fully behaves as an AHS.

It must be stressed that, on looking at Figure 1, Figure 2 and Figure 3, and from a comparative analysis between the behavior of the density correlation functions of this dendrimer–methanol mixture (at high and moderate values of φ) at all the studied temperatures, that the DWF (or the non-ergodicity factor, $f_{q}$) measured values are strongly *T* dependent. In particular, at the same volume fraction, they grow strongly as the temperature decreases (or the AHS phase develops).

Such a behavior is defined in the β-relaxation by the caging effects on density correlation decay coupled with the local thermal vibrations. Therefore, it is extremely sensitive to solute–solvent interactions from which, among other things, the attractive interaction between the colloidal nanoparticles also originate.

The essential difference in the density correlation functions decay between HS and AHS is just in their well defined clustering process similar to that which characterizes the sol-gel transition typical of percolation phenomena [53,54]. These clusters with dynamic structures and finite lifetimes are arranged as percolating clusters in a hierarchical self-similar way; and the observed logarithmic behavior originates from this precise typology of local order.

With an OH terminal group, these used dendrimers are essentially hydrophilic and therefore behave similar to AHS if they are dissolved in water (where the local order is fully dominated only by the tetrahedral network of HB); but, in a methanol solution, depending on the collective system thermodynamics (*T* and φ), the hydrophobicity originates the HE heavily influencing the physical properties of the solution on the basis of its characteristics and preventing the clustering process observed in the water solutions (Figure 4).

The results of the present study clearly indicate that the HE process is highly temperature-dependent. While, at 313 K, this colloidal system is entirely HS at 283 K and, at the same concentrations, it instead shows all the characteristic properties of an AHS. To be precise, these differences are observable both in the ISF and in the DWF non-ergodicity factor values.

All of this is coherent with the NMR results for water confined in hydrophobic nanotubes [47] and in water methanol solutions [48], suggesting the existence of a temperature induced hydrophobic–hydrophilic crossover. A reality is mirrored by the water polymorphism with the development and growth of the LDL phase, dominated by the lifetime of the HB, which grows exponentially as the temperature decreases up to the metastable undercooled state [7].

One important thing, however, needs to be clarified: while these NMR experiments investigate local molecular properties, such as the current, the PCS study, instead, deals with collective properties on a nanometric length scale. This, therefore, identifies a new broad-based investigation channel to elucidate the basic mechanisms of HE, especially in nano-systems, through the appropriate use of scattering techniques (e.g., neutron, light, X-ray in the self or distinct, coherent, or incoherent techniques).

Finally, to clarify how gradually the system under study evolves from HS to AHS (or vice versa), we consider what happens at intermediate temperatures. If we look at the ISFs’ evolution (changing *T*), we observe that the logarithmic decay begins to develop in the β-decay region at the low volume fractions, gradually evolving towards the highest until it disappears. Another significant reality, in this context, of the ISFs of these colloidal solutions in methanol, is represented by the evolution with the temperature and the volume fraction of the DWF values. The latter is essentially generated by the correlations between the nano-particles, which are mediated by the solute–solvent interactions.

When the hydrophilicity is very weak, a clustering process develops between the methyl groups; in fact, well defined structures arise, involving a vast quantity of solvent molecules. These structures created by the solvent are usually not very stable and are also influenced by the solute and its concentration.

A situation similar to that which leads to the formation of micellar structures in amphiphiles is observed [16]. Furthermore, outside these structures, there are hydroxyl groups, and a solute networking is possible. In parallel, the relative increase in the concentration of the solute with respect to that of the solvent disfavors this clustering. The temperature decrease accompanied by the development of a progressive dominance of HB with respect to hydrophobicity means that the supramolecular clustering progressively extends to higher volume fractions until the colloidal suspension becomes fully AHS.

## 3. Materials and Methods

### 3.1. Samples and Experiment

Dendrimers have a relatively homogeneous radial density with flexible end groups (with a certain degree of back folding). Whereas, the low generations g<5 are characterized by a fractal-like internal structure, and the generations with g≥5 display scattering distributions that are consistent with a homogeneous monodisperse sphere distribution [56]. Depending on *g*, their radius ranges from 5 to 60 Å and the molecular weight from some tens to $10^{6}$ Da (for g=6, $M_{W}$∼40.000 Da, and $R_{g}∼22$ Å, the same order of magnitude of small proteins such as lysozyme). From moderate integer generation, g>5, they are classified as true colloids having a hard core plus a corona region made of the flexible end groups (whose fractional depth decreases on increasing *g*). These corona flexible end groups can behave similarly to the polymers of grafted colloids, giving rise to an attractive interaction [56]. In the current experiment, we have used PAMAM dendrimers of generation g=6, with OH terminal groups, obtained from Dendritech Inc. (Midland, MI, USA) as methanol solutions. After the complete solvent evaporation (slightly and at room conditions) and the measure of the solute volume, samples were prepared at the desired φ by adding the solvent. The experiment was made for 0.08<φ<0.6 in order to observe the dynamic arrest, DA, or liquid–glass transition, accompanied by a dynamic slowing down, which marks a dramatic change in the system’s physical properties and represents a hot topic in condensed matter research.

PCS data have been taken using a digital correlator with a logarithmic sampling time scale, which allows an accurate description of both the short- and long-time regions of the intermediate scattering function (ISF) from 0.1 s to 100 s. The digital correlator converts the incoming train photo electronic pulses (detected by a single pulse photomultiplier) into a photon counting correlation function, related to the scattered intensity I(q,t) [49]. The experiments were made at the wave-vector q, corresponding ro the scattering angle of θ = 90${}^{\circ }$ using a continuous wave solid state laser (Verdi-Coherent operating a 50 mW (5120 Å) and an optical scattering cell of a diameter 1 cm in a refractive index matching bath ($\left|q\right|=4\pi n/\lambda_{0}sin(\theta /2$), *n* being the system refractive index and $\lambda_{0}$ the vacuum light wavelength. The sample is thermostated with a stability within ± 10 mK. The intensity data were corrected for turbidity and multiple scattering effects.

We worked in the homodyne mode measuring the intensity correlation function $g_{2}\left(q,t\right)=\langle I\left(q,0\right)I\left(q,t\right)\rangle /{\langle I\rangle }^{2}$ related to the dynamic structure factor S(q,t), as the scattered electric field normalized density autocorrelation function $g_{1}\left(q,t\right)=S\left(q,t\right)/S\left(q,0\right)=\langle E_{s}^{*}\left(q,0\right)E_{s}\left(q,t\right)\rangle /\langle |E_{s}{|}^{2}\rangle$ through Siegert’s relation $g_{2}\left(q,t\right)=1+{\left|g_{1}\left(q,t\right)\right|}^{2}$.

In order to measure the normalized intensity correlation function (or ISF), we used a method with the special average procedure [57]:(1)$$g_{2}\left(q,t\right)=\langle \langle I(q,t)I(q,0)\rangle /{\langle I(0)\rangle }^{2}\rangle .$$

The first bracket denotes the time average and the second the positional average over different parts of the sample. This is for the system structural arrest at high φ and the corresponding non-ergodic behavior (with the correlation functions that can develop very long time delays). Special care must be taken in averaging over many different sample positions for a correct measure of the corresponding long time regions. A large number of measurements must be performed, enough for a correct average, observing a larger scattering area by changing the sample position and exploring many scattering volumes. In such a way, for each ISF, a typical overall experimental time of about two hours is necessary in the highest φ regimes; for φ>0.3, such a procedure was repeated several times.

### 3.2. Models and Results

The ISFs, or density correlation functions $g_{1}\left(q,t\right),$ (obtained from Equation (Equation 1)), measured in the PAMAM dendrimer (OH terminal groups) of generation g=6 in $D_{2}O$ by means of the PCS technique, are proposed the Figure 4 [55]. These correlations were measured for T=293 K for a set of volume fractions within the interval 0.08<φ<0.57. Obviously, such a system is characterized exclusively by HB bonding and, therefore, by an attractive interaction between all its components (solvent–solvent, solute–solvent, and solute–solute).

Our analysis deals essentially with the MCT, able to identify in terms of power laws and universal exponents, which are functions of a control parameter (*T* or φ), an the density correlator time decays upon approaching the glass transition [52]. These power laws quantify, as in the case of critical phenomena, the fractional distance (ε) from the critical kinetic transition (at $T_{c}$ or $\varphi_{c}$) as, e.g., $\varepsilon =|T-T_{c}|/T_{c}$. In particular, the ideal MTC defines these universal characteristics of the decay (proposed in the ISFs of Figure 4), introducing the temporal behavior of the correlator in different time ranges: (a) the region of the microscopic decay at the short time $t<t_{0}$ (for colloids of the order of the Brownian motion—the only contribution present in the ISF at the lowest concentrations); (b) the β-relaxation region (cage motion), corresponding to the decay towards a plateau $f_{q}$ (Debye-Waller (or non-ergodicity factor DWF—an attenuation factor of the scattered intensity arising from local thermal vibrations))—and a further decay below the plateau (ε<0), while the ergodicity breaking take places for (ε>0); (c) and, finally, the α-decay regime represents the decay’s last stage (the cage breakup—at the highest times of the ISF). All these three regimes are well evident in the reported ISFs.

The MCT proposes, close to the plateau $f_{q}$, the following general expression for the density correlators:(2)$$F\left(q,t\right)=f_{q}+h_{q}\varepsilon^{1/2}g_{\pm }\left(t/t_{\varepsilon }\right).$$
where the subscript in the β-correlator ($g_{\pm }$) corresponds to the sign of ε. Hence, F(q,t) has separate space and time dependences and is characterized by the time scaling $t_{\varepsilon }=t_{0}/{\left|\varepsilon \right|}^{1/2a}$. In these terms, the correlator obeys precise time scale laws before and after the plateau, hence: (i) for $t/t_{\varepsilon }<<1$, it is valid the temporal scaling law $g_{\pm }\left(t/t_{\varepsilon }\right)\approx {\left(t/t_{\varepsilon }\right)}^{-a}$; while, for $t/t_{\varepsilon }>>1$, the system dynamics obey von Schweidler’s law,
(3)$$F\left(q,t\right)=f_{q}+h_{q}{\left(t/t_{\varepsilon }^{{}^{\prime }}\right)}^{b}.$$
which includes a second characteristic time scale $t_{\varepsilon }^{{}^{\prime }}=t_{0}/{\left|\varepsilon \right|}^{\gamma }$ with $\gamma ={\left(2a\right)}^{-1}+{\left(2b\right)}^{-1}$ and $h_{q}$ as a constant amplitude. Unlike the exponents characterizing the critical transitions, a and b are non-universal quantities that depend only on the so-called exponent parameter λ; these are determined by the static strucure factor S(q). In particular is $\lambda =\Gamma^{2}\left(1-a\right)/\Gamma \left(1-2a\right)=\Gamma^{2}\left(1+b\right)/\Gamma \left(1+2b\right)$, Γ, being the Euler gamma function with 0<a<0.5 and 0<b<1. Finally, we mention that the the α-decay regime is described by means of the strectched exponential function $\phi \left(q,t\right)=A_{q}exp\left[-{\left(t/\tau_{q}\right)}^{\alpha }\right]$.

From the Figure 4 the MCT DWF can be observed (indicated as $f_{q}$ and represented by the horizontal line). In the liquid state (ε<0), at the lowest φ, this long-time limit of ISF (α-decay) assumes moderate values, while it grows up to values $f_{q}>0.9$ for φ>0.5 by increasing the volume fraction. The structural arrest transition is characterized by a $f_{q}$ discontinuous change called a bifurcation transition; the ISF measured at about 0.547 thus shows the predicted liquid-to-glass transition.

Just the behavior of this last ISF (φ=0.547) can be used to illustrate the different temporal regions typical of MCT and the way in which the corresponding relaxations can be evaluated (see e.g., Figure 5). For this reason, by means of some vertical lines, we have divided the relative temporal domain into five regions indicated by Roman numerals, from I to V. The two outer regions, respectively, represent: I, the microscopic motion (the fast intra-cage dynamics—for this reason the ISF corresponding to the lowest concentration is also reported), and V, the slow inter-cage dynamics (the α decay regime). II, III and IV, instead, represent the time regions characteristic of the β-caging dynamics (power laws): i.e., the decay toward the plateau, the plateau where there is the non-ergodicity onset, and the final region (higher times) described by the von Schweidler’s law, respectively. The dashed line represents a fit of the this latter scaling law (Equation (Equation 3)), which gives as a result λ=0.7 and γ=2.3; this illustrates both the goodness of the MCT scaling laws and the way in which relaxation times are measured.

As can be easily observed from Figure 4, an important characteristic emerges in the reported density correlations: the occurrence of a region of the logarithmic time dependence, located in the system ergodic state preceding the plateau (pointed out by the superimposed straight lines in the linear-log plot). This is the signature of a liquid-to-attractive glass transition. This situation is illustrated in Figure 6 through a zoom of the ISFs preceding the dynamic arrest (0.21<φ<0.547). The lines are the ISF data fits in the temporal region of this logarithmic decay. More precisely, these data show that this special and new structural relaxation takes place inside the region of beta caging dynamics.

As we will discuss later, such a ISF logarithmic behavior is the central part of our study because it is due to the competition between the HE and HB interactions. Its effects will be verified in the study of the same dendrimer in solution with the methanol (a substance which has a hydrophilic and a hydrophobic group) by considering the evolution of the measured ISFs (at different φ and *T*).

Figure 7 illustrates the ISFs corresponding to the same dendrimer colloidal solution previously proposed (T=293 K and 0.11<φ<0.6 ), but with the methanol as solvent. A substantial difference emerges from the previous case (Figure 4) where the solvent was water: (a) at the same *T*, and at all the used φ, only some of the illustrated ISFs seem to show the AHS logarithmic decay (0.37<φ<0.39), and (b) the resulting DWF values are considerably lower than those previously measured. This is due to the action of the solvent hydrophobicity on the solute (dendrimer with only OH terminal bonds). Figure 8, instead, shows the measured ISTs for the same colloidal solution (and same concentrations) for the highest studied temperature T=313 K. For all the φ investigated, none of the ISFs show the AHS logarithmic decay: conditions for which the interactions between the dendrimers appear to be dominated by a repulsive hydrophobicity.

### 3.3. Data Analysis

The ideal hard spheres’ MCTs (where the explanation of β relaxations fully satisfies the so-called von Schweidler scaling relations) were developed originally by using cage effects, determining the relaxation time dependence [52]. At very short times, density relaxation reflects the localized motion of individual particles, entailing the details of the hydrodynamic interactions. At longer times, particles are instead trapped in their neighbor structure. At $\varphi_{c}$, both the particle diffusion and the long-time density fluctuations freeze, and the system undergoes an ergodic to non-ergodic transition, defined as the kinetic glass transition.

These ideal MCT predictions have been successfully tested, both experimentally and by using MD simulations, for various systems. In particular, this was also performed for a suspension of PAMAM dendrimers (PAMAM $NH_{2}$ (g5)) in methanol at different concentration. In this latter case, a complete agreement with the β-decay von Schweidler scaling relations and the corresponding exponents has been observed in the region 0.373<φ<0.518) [53].

The MCT in the extended form showed additional relevant predictive power laws in the field of glassy system dominated by attractive interactions, where new interesting phenomena have been discovered. Regarding the original MCT in the ideal form, it must be stressed that the predicted glass line usually differs greatly from the locus where the arrest (or glass transition) is observed experimentally (the predicted $\varphi_{c}$ is 0.516, whereas the measured one is 0.58 [58]). The reason for this mismatch was ascribed to activated processes not considered in the ideal MCT.

MCT calculations with a proper description of the particle interactions [59,60] show that, if the system is characterized by a hard core plus, an additional short-range attractive interaction (an adhesive hard-sphere system AHS), as well as a different arrest scenario with two types of structurally arrested (glassy) states, emerge. In this case, a second external control parameter, temperature, must be introduced into the description of the phase behavior, in addition to φ; loss of ergodicity can take place by either increasing φ or changing the *T*. At high *T* and at sufficiently high φ, the system evolves into the well known hard-sphere repulsive glass. However, at relatively low *T*, an attractive glass can form, where the particle motion is hindered by cluster formation with neighboring particles. In an AHS, aside from the hard-core diameter, an additional length scale, the range of the attractive well, should come into play.

According to this picture, we may divide spherical colloids into two categories: the one-length scale hard-sphere system, in which the glass formation is dictated by the cage effect, and the two-length scale AHS, in which the two glass-forming mechanisms may coexist and compete. Furthermore, MCT shows that, in an AHS with sufficiently small ratios of the range of the attractive interaction to the hard-core diameter, variation of the control parameters allows the transition between these two distinct forms of glass.

Of special interest is the occurrence of an A3 singularity where the glass-to-glass transition line terminates [52]. MCT suggests that the two distinct dynamically arrested states become identical at and beyond this point.

The short range attraction also has as a consequence the logarithmic decay of the temporal relaxation in the ergodic liquid state [54,61]. Here, the density correlation functions exhibit such a peculiar time dependence followed by the α power-law decay. The combination of neutron scattering and PCS measurements have given evidence of this physical reality in a copolymer micellar system [53,62]. The Figure 4 ISFs illustrate that the range 0.21<φ<0.50 corresponds to the AHS ergodic liquid state, as proposed by the different straight line. At high φ, the system evolves into the glassy state with the corresponding $\varphi_{c}≃0.54$ (a value in agreement with the theory).

The ideal MCT limits have been overcome through the development of a version of the model that incorporates activated hopping processes: the extended MCT [63,64]. Such a model, for a general structure of its equations of motion, predicts a crossover in the β -relaxation time, just near the critical temperature $T_{c}$ of the ideal version. All of this is in accordance with the observation of a universal fragile-to-strong dynamic crossover (FSDC) in which the glass forming liquids transport functions change from a super-Arrhenius to an Arrhenius behavior at $T_{c}>T_{g}$. This FSDC is accompanied by the violation of the Stokes-Einstein relations [65].

At this point, we want to underline that the present study is not dedicated to an extensive and quantitative analysis of MCT and its models (fully tested), but through the use of some of its universality, it verifies the behaviors of HE, especially as a function of temperature using nanoparticles such as dendrimers. In particular, we will use this special and exclusive feature of AHS in its liquid phase: the relaxations characterized by specific logarithmic behaviors.

## 4. Conclusions

By using the MCT extended model, we have evaluated, from the ISFs (density correlation) measured by means of PCS experiments, the competition between the hydrophilicity and hydrophobicity (hydrophobic effect) in dendrimer colloidal solutions (with OH terminal groups) in water and methanol. To clarify this intriguing situation, we have studied many different volume fractions, from diluted ones up to higher values, leading to the glass transition dynamic arrest ($\varphi_{c}$). While, for the solutions in water, we worked at a fixed temperature (293 K), and, for those in methanol, we have evaluated what happens by changing *T* in the range 283<T<313 K.

The datum point of our analysis is in the strong difference, as proposed by MCT, characterizing the ISFs of hard-spheres from the attractive colloidal systems; the difference is characterized, for these latter macrostructures, by the presence of the logarithmic time dependence in the β-relaxation preceding the plateau region in the ergodic state just before the critical transition. This is a behavior entirely due to the short range attraction. In fact, in the case that the solvent is water, the density correlation functions show such a characteristic relaxation for all the volume fraction below the dynamic arrest.

Instead, the behavior of the same dendrimer dissolved in methanol is very different: such an AHS decay is strongly dependent on both concentration and temperature. Regarding this latter dependence, while at the lowest investigated *T* (283 K), this characteristic decay is (as in the case of the solutions in D_2_O) present at all concentrations of the liquid phase, and, at the highest (313 K), it is always absent. So, in an interval of about 30 K, on decreasing *T*, such NP suspension evolves from a purely HS repulsive behavior dominated by hydrophobicity to one whose interaction properties are entirely hydrophilic, with the HB playing the fundamental role.

Between these two extreme temperatures, the logarithmic decay is observable at the low volume fractions whose range increases as the temperature of complete hydrophilicity is approached (283 K).

In addition to this, for the same solute, a significant evolution is observed between the measured values of the Debye Waller factor (the non-ergodicity temperature factor, $f_{q}$), strongly dependent on the solvent as well as on the composition, but, above all, also on the temperature. As is well known, such a physical quantity is generated exclusively by molecular thermal effects. It is, in fact, in our case (MCT), an “attenuation” factor of the measured intensity of the dynamical structure factor (i.e., the Fourier transform of the $g_{1}\left(q,t\right)$) arising essentially from local thermal intra-cage vibrations.

This last quantity, in addition to the β-decay, is strongly associated with the effects of the competition between the hydrophilic and hydrophobic molecular tails. It is the HE that determines the dynamics (and its thermodynamic evolution) of the nanostructures in both solutions, water or methanol.

A thermodynamic behavior in these nano-system solutions provides an accurate description of how the hydrophobic effect (HE) develops in agreement with the findings of the previous NMR experimental studies on water confined in hydrophobic nanotubes and water–methanol solutions [47,48]. The hydrophobic–hydrophilic crossover depends on the temperature; in particular, at temperatures higher than ambient, the hydrophobicity becomes gradually stronger and governs the system interactions while, at lower ones, HB dominates. However, our data show a dependence of this effect also on the volume fraction of the colloidal solution at intermediate temperatures (see e.g., Figure 8), indicating that analogous effects can be verified with greater accuracy by studying the system with the PCS experimental technique (or neutron scattering), the MCT models, and the same nanoparticles dispersed in solutions of water and methanol at different molar fractions.

The presented data detail the temperature (and concentration) dependence of the hydrophobic effect—i.e., the competition between hydrophilic and hydrophobic interaction—at the nanoscale, extending the results of other experiments that showed it at the microscopic level [47,48]. Stressing again that while the former is characterized by a well known thermodynamic widely detailed in the literature, the second instead (the hydrophobic one) still lacks an equally detailed knowledge even at an analytical level.

We close by considering that, unlike the previous experimental studies focused on molecular (and therefore local) properties of the system, the current experiment instead concerns the consequences of HB, and hydrophobicity, on the collective ones, i.e., how the competition between these two opposite contributions determine the interaction between the different nanoparticles

The results obtained, using a fundamental model of statistical physics of dense glass forming systems (MCT) together with the experimental method of scattering, clearly suggest that, on this basis, we can improve our knowledge on the interactions that characterize nano-materials.

## Figures and Tables

**Figure 1 ijms-24-02003-f001:**
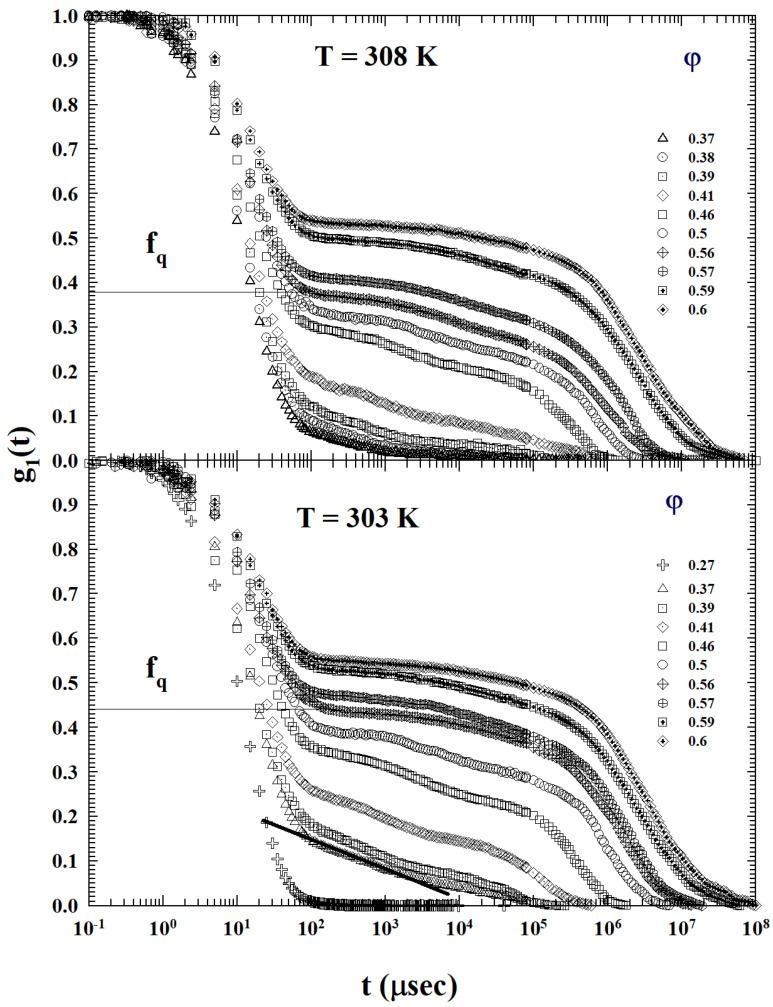
The dendrimer methanol ISFs measured in the range 0.27<φ<0.6 for T=308 and 303 K. For T=308 K (**top**) the behavior are analogous to that at T=313 K (figure in Section 3.2): none of the illustrated ISFs show the AHS logarithmic decay. Whereas, for 303 K (**bottom**), such a decay is observable only for φ=0.37. In both cases, the critical ($\varphi_{c}$) appears to be greater than φ=0.5.

**Figure 2 ijms-24-02003-f002:**
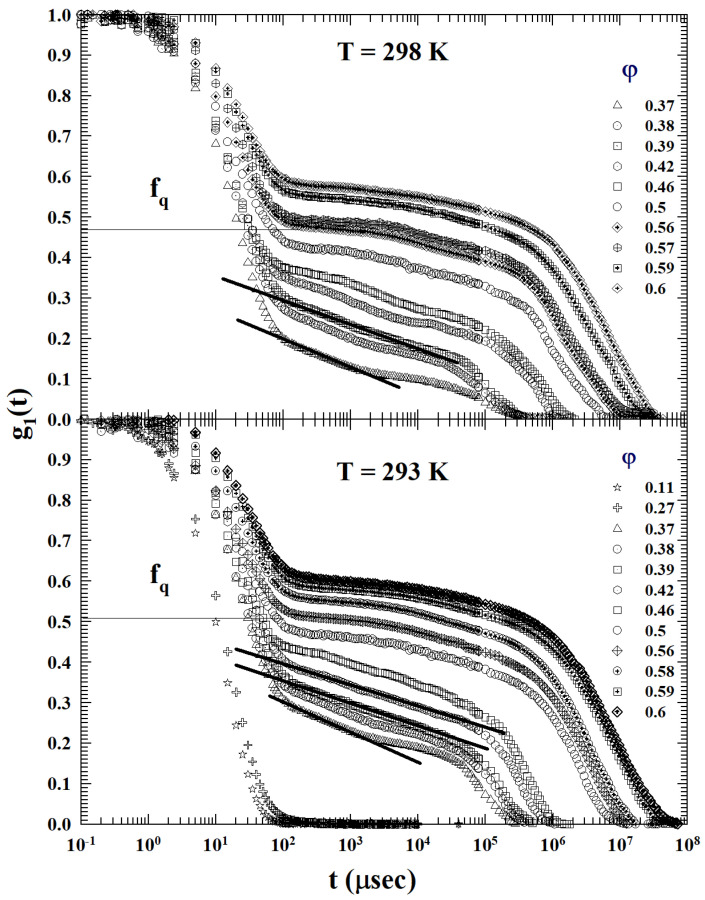
The dendrimer-methanol ISFs, at the different φ for T=298 and 293 K show that a *T* decrease leads to an increase of the concentration range in which the logarithmic decay exists. In the first case, the decay typical of AHS systems can be observable for 0.37<φ<0.39. Instead, for T=298 K, the corresponding φ range is 0.37–0.42; also, here we have $\varphi_{c}>0.5$.

**Figure 3 ijms-24-02003-f003:**
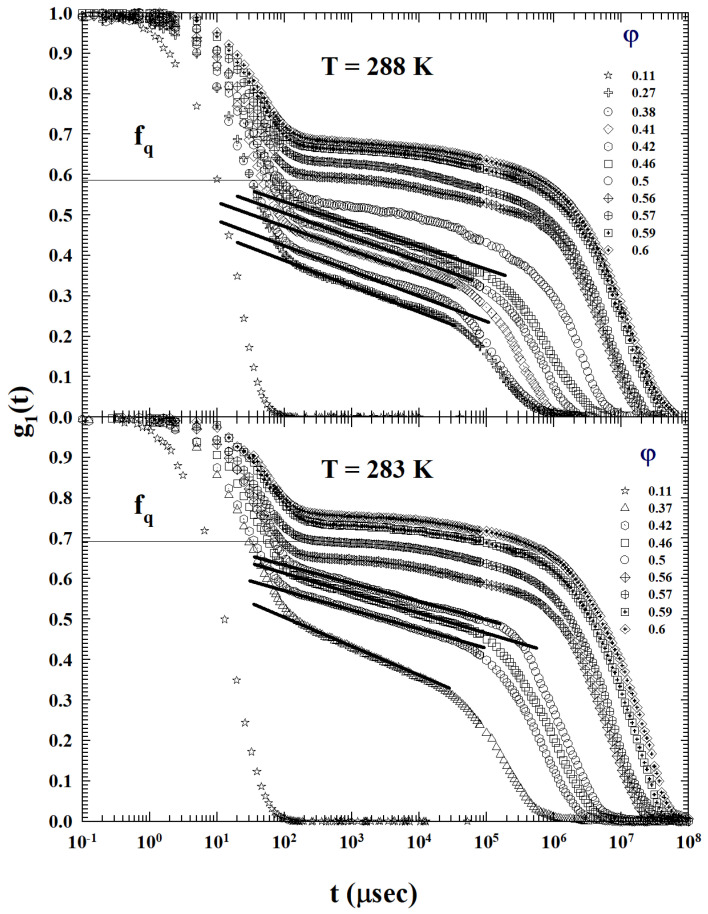
The φ behaviors of the dendrimer–methanol ISFs at the two lowest studied temperatures (T=288 and 283 K). As can be clearly seen at 283 K, the logarithmic decay is present in all the liquid states up to $\varphi_{c}$, indicating that the system behaves fully as an AHS.

**Figure 4 ijms-24-02003-f004:**
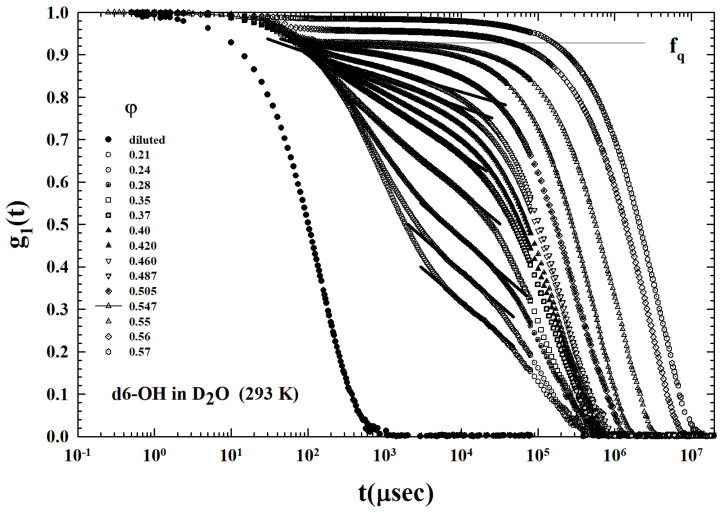
The intermediate scattering functions (ISF), measured at T=293 K for different volume fractions (φ, in the range 0.1–0.57) in the $D_{2}O$ solutions of PAMAM dendrimer (generation g=6), with OH terminal groups. The logarithmic decay (straight lines) typical of adhesive hard spheres (AHS) colloids can be observed for all the φs of the liquid region before the arrest at about φ=0.547. $f_{q}$ denotes the Debye-Waller factor (DWF). Figure adapted from Ref. [55], Copyright (2014) Elsevier.

**Figure 5 ijms-24-02003-f005:**
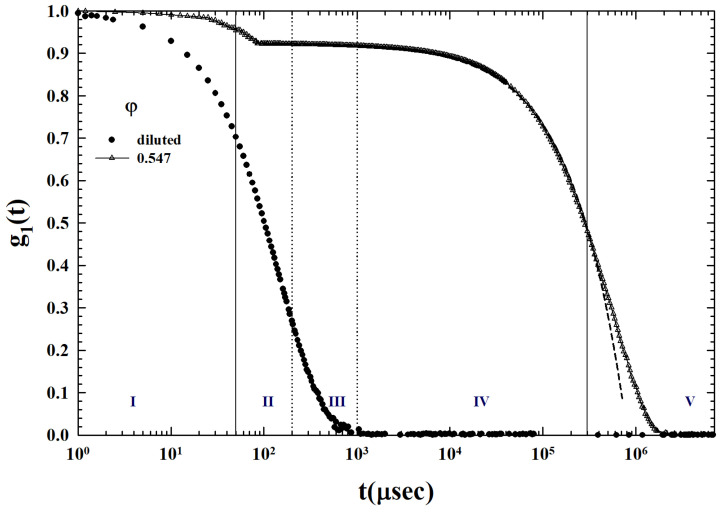
The ISFs for φ=0.547 and the diluted one, together with the different MCT time domains from the microscopic motion (I) to the slow inter-cage (V), and the the β-caging dynamics (II, III and IV). The dashed line is a fit of the von Schweidler’s law scaling law.

**Figure 6 ijms-24-02003-f006:**
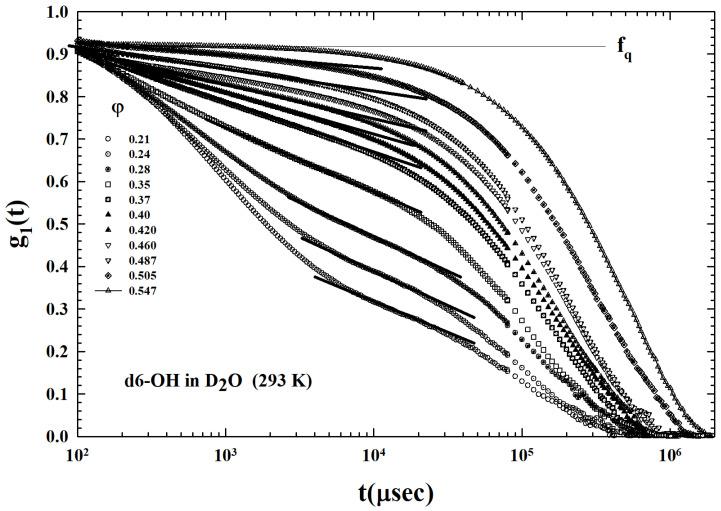
A zoom of the Figure 4 ISFs preceding the dynamic arrest (0.21<φ<0.547). The lines are the ISF data fits in the temporal region of the beta caging dynamics showing the logarithmic decay.

**Figure 7 ijms-24-02003-f007:**
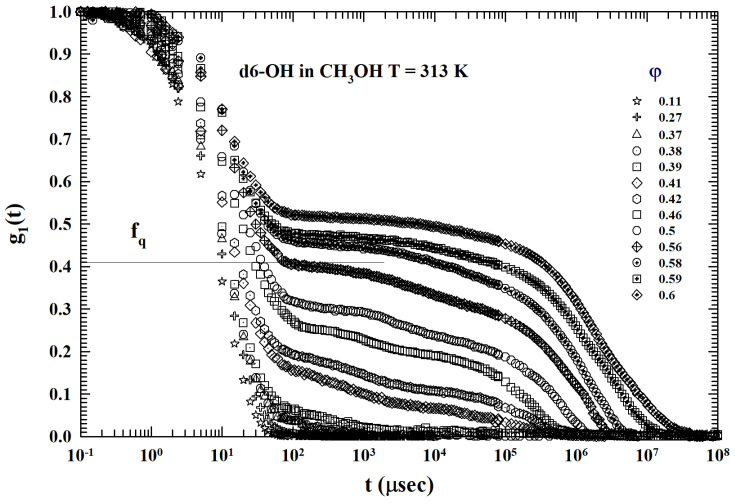
The ISFs of the g=6 dendrimer dissolved in methanol at 313 K. In this case, none of the illustrated ISFs show the AHS logarithmic decay. Thus, the system appears to be fully dominated by hydrophobicity.

**Figure 8 ijms-24-02003-f008:**
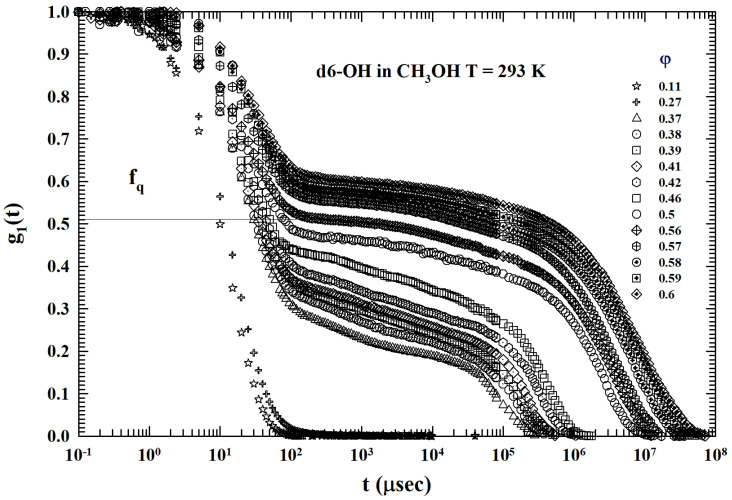
The ISFs of dendrimer (the same of Figure 4) in methanol for 0.11<φ<0.6. In the actual case (water as solvent): at the same *T* and at all the used φ, only some of the reported density correlations seem to show the AHS logarithmic decay (0.37<φ<0.39). In addition, the resulting DWFs are considerably lower than that measured previously. This is the effect of the solvent hydrophobicity on the solute (dendrimer with only OH terminal bonds).

## Data Availability

The data that support the findings of this study are available from the corresponding author upon reasonable request.

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
