# Peer review of "The Hydrophobic Effect Studied by Using Interacting Colloidal Suspensions"

_ijms, 2023, doi:10.3390/ijms24032003_

Round 1

Reviewer 1 Report

This paper by Mallamace et al. discusses photon correlation spectroscopy (PCS) experiments of hydroxylated dendrimers in methanol. The intermediate scattering functions (ISF) of these systems have been measured as a function of colloid concentration and temperature. The results are analyzed using concepts of Mode Coupling Theory (MCT), which describes the relaxations of a system that evolves towards a phase transition to a glass. I have found that the paper contains original results and an expert analysis and discussion, so the paper should be published, but it requires some revision to increase its readability. There are some points in the paper, which I suggest must be fixed to make it more readable for a broader audience. The paper has been written strictly for the community of researchers in the field of statistical physics of liquids. As such, it will not be easy to read by nonexperts in this area, as it was not easy for me. The introduction is supposed to offer an introduction to the field, but it is amazingly long (6-7 pages) and tedious and thus of little help. Since the paper most probably will not be read by nonexperts, I suggest that the introduction should be strongly reduced to the minimum required. I do not see the need for the material in lines 48-70, 106-117, 154-165, 199-203. Figure 1 is completely irrelevant and should be removed. It is a nice picture, but does not connect to the topic investigated. Instead, the introduction should be terminated with a couple of more transparent paragraphs, in which the reader would find precisely what is attempted in the paper, and what is the real innovation. The information exists in various parts of the lengthy introduction, but in a very scattered way.

I found the ISF figures very hard to follow, as they are full of experimental curves at different volume fractions. Improvement of the figures can be achieved by presenting fewer points in each curve and enlarging the points a bit. I am not very familiar with this type of analysis, but I wonder if the characteristic relaxation times in these systems could be further tabulated and analyzed.

Finally, since my background is in more traditional physical chemistry, I fail to see how the authors envision the hydrophobic effect. They keep talking of hydrophobic interactions, and they complain about the fact that there is no clear model and understanding of these interactions. Yet, in lines 224-229 they refer to a hydrophobic effect potential (VHE), on the same footing as the Coulomb or Van der Waals potential, or even a potential that can be ascribed to hydrogen bonds. I am quite familiar with the latter potentials, but not with any potential VHE that can model the hydrophobic effect per se. Does not the hydrophobic effect contain entropy components?

Author Response

First of all, we feel the duty to thank you for the work he has done and the proposed suggestions to improve the submitted paper; hence we have revised the manuscript in full accordance with the (minor) corrections requested.

In particular, we have eliminated Figure 1 and revised the English form.

In addition, we have completely revised the introduction chapter by eliminating all the redundant (and of little help to the reader) parts, thus reducing it from 7 to 5 pages. We therefore believe that the current form is significantly clearer and more coherent.

Finally, we have reconsidered the ISF figures in order to make them clearer and easier to interpret. With regard to a possible analysis of the characteristic relaxation times evidenced in the related ISFs, we have stressed that such a study has previously been done, as reported in reference 76, in terms of the MCT scaling laws (the von Schweidler relations for the (PAMAM NH2 dendrimer (g5) in methanol, at different concentration) obtaining a universal behavior and values of the characteristic exponents in agreement with the theoretical predictions.

Reviewer 2 Report

The authors perform photon correlation spectroscopy on dendrimer colloids that are functionalized by OH groups and thus water soluble. The authors show the raw data in six figures at different temperatures and volume fractions. The very lengthy introduction suggests that the interplay of hydrophobic and hydrophilic interactions is studied by the experiments, but from reading the results section I could not understand what the authors are trying to show by the data. The discussion of the data is incomprehensible to me, I also did not understand what is learned by comparing experiments in water and methanol. The authors mention Mode-Coupling Theory a few times, but the predictions of Mode-Coupling Theory are not quantitatively compared with experiments. I have no idea why figure 1, the picture of a flower, is shown, this is completely unrelated to the data. This paper cannot be published.

  1.  
  2.  

Author Response

Although we feel obliged to thank you for the work done, we believe that unfortunately no dialogue can be activated between us. The reason is that we are afraid that you probably don't have the right familiarity with the main themes of the manuscript, i.e. hydrophobic effect and the Mode-Coupling theory.

Reviewer 3 Report

Paper entitled “The hydrophobic effect studied by using interacting colloidal suspensions” coauthored by Francesco Mallamace, Giuseppe Mensitier, Martina Salzano de Lun  and Domenico Mallamace represents a photon correlation spectroscopy study of dendrimer colloids with OH terminal groups dissolved in D2O and methanol. The work was pointed to exploration of the hydrophobic effect versus hydrophilicity on temperature and the solute volume fraction. The interpretation of density correlation functions obtained from the intermediate scattering functions was based on the Mode Coupling Theory applied to models of hard-fully repulsive spheres and attractive colloids. The indications of the liquid-to-attractive glass transition were interpreted on different conditions of temperature, solvent (water/methanol) and solute volume fraction. It was shown that in some range of the volume fraction the dendrimer colloids with hydrophilic terminal groups in methanol undergo the glass transition like attractive hard spheres and above that concentration the glass transition is typical for repulsive colloids. This range of fraction gets wider at the decrease of temperature. One of the conclusions is the strong dependence of the hydrophobic effects on temperature which leads to attractive sphere behavior at lower temperatures, whereas spheres are repulsive at higher temperatures. That result is in agreement with the previous NMR studies but now the hydrophobic effect is studied as averaged on the scale of colloid particles.

   The paper looks like written in a hurry and requires corrections in respect to the English grammar. For example:

line 64: "could can"

line 120: "the s" the sum ?

line 135: structure ... depend ...

line 156: "...or the high tides on the beach to the ion channels in cell membranes"

line 166: "the HE and ... is qualitatively described"

line 260: "... large tile molecules..." ("... large tail molecules..."?)

etc.

   The dividing introduction into the two sections and figure 1 raises questions. Such a picture would be relevant for a paper in a popular journal such as the Scientific American.  

   More serious concern is about the main conclusions. The temperature dependence of the hydrophobic interactions is rather evident, as they originate from spatial arrangement of solvent molecules at the interface. There is nothing exciting in extension of this conclusion from molecular to nano-scale level: this conclusion is verified even on the macroscopic level (the dependence of the surface tension of liquids on temperature). The result of the study is rather methodology of applying the mode coupling theory to dendrimer colloids.

Author Response

First of all, we thank you for the work he has done and the proposed suggestions to improve the submitted paper; hence we have revised the manuscript in full accordance with the (minor) corrections requested.

In particular, we have eliminated Figure 1 and revised the English form.

Finally, as suggested, we have reconsidered the conclusions clarifying in detail what he underlined.

Round 2

Reviewer 2 Report

The authors did not intend to respond to my previous report, I therefore maintain my previous judgement and to recommend publication.

Author Response

(The authors gave the same response as above.)
